# Poor air quality is associated with impaired visual cognition in the first two years of life: A longitudinal investigation

John P Spencer[1]*, Samuel H Forbes[2], Sophie Naylor[1], Vinay P Singh[3], Kiara Jackson[1], Sean Deoni[4], Madhuri Tiwari[3], Aarti Kumar[3]

[1]School of Psychology, University of East Anglia, Norwich, United Kingdom; [2]Department of Psychology, Durham University, Durham, United Kingdom; [3]Community Empowerment Lab, Lucknow, India; [4]Department of Pediatrics, Brown University, Providence, United States

## Abstract

**Background:** Poor air quality has been linked to cognitive deficits in children, but this relationship has not been examined in the first year of life when brain growth is at its peak.

**Methods:** We measured in-home air quality focusing on particulate matter with diameter of <2.5 μm ($PM_{2.5}$) and infants' cognition longitudinally in a sample of families from rural India.

**Results:** Air quality was poorer in homes that used solid cooking materials. Infants from homes with poorer air quality showed lower visual working memory scores at 6 and 9 months of age and slower visual processing speed from 6 to 21 months when controlling for family socio-economic status.

**Conclusions:** Thus, poor air quality is associated with impaired visual cognition in the first two years of life, consistent with animal studies of early brain development. We demonstrate for the first time an association between air quality and cognition in the first year of life using direct measures of in-home air quality and looking-based measures of cognition. Because indoor air quality was linked to cooking materials in the home, our findings suggest that efforts to reduce cooking emissions should be a key target for intervention.

**Funding:** Bill & Melinda Gates Foundation grant OPP1164153.

## Editor's evaluation

This study presents an important finding on the negative association of indoor air quality with visual cognition in the first two years of life. Key strengths include the longitudinal design, fine-grained measures of indoor air quality, and multi-modal assessment of cognitive functioning in a large sample of infants from families across diverse SES strata. The evidence provided is solid and will be of interest to researchers working in the fields of neurocognition, child development, environmental and public health.

## Introduction

The impact of poor air quality on neurocognitive health is a global concern. This impact was recently quantified by the Global Burden of Disease Study in India, attributing 1.67 million deaths in 2019 to air pollution-related causes with an overall national economic loss of $36.8 billion (*Pandey et al., 2021*). Such economic losses are compounded by evidence that poor air quality impacts neurocognitive health in childhood as economic losses accumulate over years of reduced productivity (*Heckman and Mosso, 2014*). Studies have reported that poor air quality and proximity to roadways is associated

*For correspondence:
j.spencer@uea.ac.uk

with reduced general cognitive functioning in childhood (*Freire et al., 2010*; *Wang et al., 2009*; *Harris et al., 2015*; *Suglia et al., 2008*) and slower growth in working memory (*Sunyer et al., 2015*). Exposure to poor air quality is also a risk factor for child emotional and behavioural problems which can have severe impacts on families (*Midouhas et al., 2019*).

But how early do these effects emerge? This is an important question as animal studies show profound impacts on the brain in early development, with inhalation of diesel exhaust and ultrafine particles resulting in elevated cytokine expression and oxidative stress in the brain (*Gerlofs-Nijland et al., 2010*; *Bos et al., 2012*), as well as altered neurogenesis (*Patten et al., 2020*). Very small particulate fragments (particulate matter with diameter of <2.5 μm; $PM_{2.5}$) are of major concern as they can move from the respiratory tract into the circulatory system reaching the brain. The brain may also be particularly sensitive during infancy due an immature detoxification response (*Grandjean and Landrigan, 2014*). Several large-scale studies have looked at the effects of prenatal exposure to nitrous oxide and $PM_{2.5}$ on human cognition early in development (*Guxens et al., 2012*; *Guxens et al., 2014*). Results show reduced psychomotor functioning at 1–6 years of age, but no associations with early cognition, although three studies have reported slower cognitive growth and emotional / conduct problems in children exposed to poorer *indoor* air quality as assessed via questionnaires (*Midouhas et al., 2019*; *Gonzalez-Casanova et al., 2018*; *Vrijheid et al., 2012*).

Critically, no studies have looked at the relationship between poor air quality and cognition in the first year of life when brain size doubles and may be particularly sensitive to toxins. This may reflect the challenge of assessing cognition in infancy. While standardized measures exist (e.g. Mullen Scales of Early Learning), these tools may not generalize to non-Western cultures where air quality is poorest. An alternative is to measure infant cognition using specially-designed looking-based tasks. Multiple aspects of visual cognition that are predictive of later cognitive abilities (*Rose et al., 2012*) can be measured reliably in the first year across cultures (*Rose, 1994*; *Wijeakumar et al., 2019*). For instance, visual processing speed measured using looking-based tasks in both infancy and toddler-hood is significantly predictive of working memory and executive function scores at 11 years (*Rose et al., 2012*). A second challenge is that infants spend much of the first year indoors. Consequently, data from outdoor monitoring stations may not accurately reflect air quality exposure critical for early brain development, particularly in contexts where use of, for instance, solid fuels in cooking may lead to differences between indoor and outdoor air quality. Recent advances in air quality monitoring allow for the measurement of $PM_{2.5}$ directly in homes; this may be critical as $PM_{2.5}$ is possibly the most neurotoxic component of residential air quality (*Block and Calderón-Garcidueñas, 2009*).

Here, we examined the relationship between poor air quality and cognition in infancy using looking-based measures of cognition and in-home measures of $PM_{2.5}$. To assess visual cognition, we used a specially-designed, transculturally-relevant visual cognition task which assessed infants' visual working memory and processing speed (Figure 1A and B; see *Ross-Sheehy et al., 2003*), capitalizing on infants' tendency to look away from visual familiarity and toward visual novelty. This task has been examined in several prior studies (*Oakes et al., 2006*; *Oakes et al., 2009*), but this is the first study to use this task longitudinally with a sample of non-western infants. Infants visually explored two displays that blinked 'on' and 'off'. On one side – the 'no change' side – the same colored squares were always presented; on the other side, one randomly-selected square changed color after each blink. If infants begin looking at the 'no change' side and they can remember the colors in working memory, they should lose interest in this display, releasing fixation to visually explore the 'changing' display. Here, they should detect the novel color and sustain looking to this display, leading to a strong *change preference* – a high proportion of looking to the changing side. This change preference is modulated by visual working memory capacity. When the number of items on the screen is small (i.e. the memory load is low), infants can detect the difference between the 'no change' and 'change' displays, and they show a higher change preference. With increasing memory load (and increasing difficulty), infants may falsely detect changes on the 'no change side' and have difficulty releasing fixation, leading to a lower change preference (*Perone et al., 2011*). Thus, the change preference score yields a quantifiable measure of infants' visual working memory abilities. Note that if infants begin looking at the 'change' side, they should remain looking at this display with a consistently high change preference score (see Methods for further discussion).

Our study was conducted in Shivgarh, India, a rural community in Uttar Pradesh, one of the states in India that has been most strongly impacted by poor air quality (*Pandey et al., 2021*). We report data

**Table 1.** Proportional distribution of key demographic indices for families classified as high versus low SES based on a median split of family SES measured using the modified Kuppuswamy scale.

| Electricity? | Income* | Cooking Fuel | High SES | Low SES |
|---|---|---|---|---|
| No | Low | Cow dung | 0.02 | 0.06 |
| | | Wood | 0.02 | 0.25 |
| | | LPG | 0.01 | 0.01 |
| | Medium | Cow dung | 0.01 | 0.02 |
| | | Wood | 0.04 | 0.19 |
| | | LPG | 0.00 | 0.00 |
| | High | Cow dung | 0.00 | 0.00 |
| | | Wood | 0.03 | 0.08 |
| | | LPG | 0.01 | 0.00 |
| Yes | Low | Cow dung | 0.04 | 0.01 |
| | | Wood | 0.08 | 0.10 |
| | | LPG | 0.04 | 0.01 |
| | Medium | Cow dung | 0.03 | 0.02 |
| | | Wood | 0.17 | 0.17 |
| | | LPG | 0.02 | 0.00 |
| | High | Cow dung | 0.02 | 0.01 |
| | | Wood | 0.22 | 0.06 |
| | | LPG | 0.26 | 0.02 |

*Incomes ranged from ₹8000 to ₹480,000 with tertile divisions at ₹45,000 and ₹72,720.

from 215 families from a range of socio-economic backgrounds (see *Table 1*). Infants were enrolled at 6 months (N=108) or 9 months of age (N=107) at which time they completed an in-lab assessment of visual cognition (see Figure 1A) as well as standardized psychomotor and cognitive assessments. A similar assessment was repeated a year later when the same infants were 18 or 21 months of age (see Methods). We examined these two cohorts based on evidence that there is an improvement in visual working memory capacity between 6 and 8 months. In particular, Ross-Sheehy and colleagues (*Ross-Sheehy et al., 2003*) showed that 6.5-month-old infants demonstrate greater-than-chance change preference scores with a memory load of one item, while 10- and 13-month-old infants showed greater-than-chance change preference scores for memory loads of two and three items. Similarly, 6.5-month-old infants could remember one spatial location, while 8- and 12.5-month-old infants showed evidence of remembering multiple locations (*Oakes et al., 2011*). By assessing infants in rural India on either side of this transition, we hoped to either replicate a similar transition or, alternatively, to reveal a delay in this transition across cultures.

## Methods
### Participants
Infants born to mothers from Shivgarh, Uttar Pradesh, India and who were aged 6 months±15 days or 9 months±15 days were eligible for participation. Infants were initially screened as belonging to either 'high socioeconomic status (SES)' (both parents having >10 years of education) or 'low SES' families (both parents have ≤ 5 years of education; see *Ahmed et al., 2018* for a similar screening approach). Infants born to parents screened with colour vision deficits (due to the nature of the VWM task), or with any congenital problems, or gestational age <26 weeks at birth, were excluded from the study.

We brought 257 families to the lab for the VWM and cognitive assessment; however, 17 families (10 6-month-old infants; 7 9-month-old infants) did not complete the assessment and were dropped from the study. The remaining 240 families were followed up for the duration of the study which spanned two years and included the following: (1) a laboratory assessment in year 1 at 6 or 9 months of age; (2) a home visit every three months thereafter for the remainder of year 1 (e.g. at 9, 12, and 15 months of age for the 6 month cohort); (3) a laboratory assessment in year 2 at 18 or 21 months of age; (4) a home visit every three months thereafter for the remainder of year 2 (e.g. at 21, 24, and 27 months of age for the 6-month cohort). Enrollment was distributed over 4 waves separated by 3 months, such that we enrolled approximately 60 infants per wave of data collection (for a full list of data collected in the study, see Supplementary Materials). The study was approved by the Community Empowerment Lab Institutional Ethics Committee (Ref. No: CEL/2018005). Participants' caregivers provided written informed consent; where caregivers were illiterate, a witness gave signed consent accompanied by a thumb impression of the caregiver in place of a signature. At the end of each laboratory session, families received a small token of appreciation.

Air quality data were collected for a total of 219 children; of those, 215 had data that survived initial data quality checks (described below). 213 children also had data from the visual working memory task and cognitive assessment during either the first laboratory assessment (when infants were 6 or 9 months of age) or the second laboratory assessment (when infants were 18 or 21 months of age). In particular, 204 participants contributed visual working memory data in year 1 when they were 6 months of age (N=109; *N* girls = 50; *M* age = 182.4 days, *SD* age = 14.9 days) or 9 months of age (N=95; *N* girls = 51; *M* age = 267.2 days, *SD* age = 14.2 days). In year 2, 181 participants contributed visual working memory data when the infants were 18 months of age (N=90; *N* girls = 45; *M* age = 546.2 days, *SD* age = 19.2 days) or 21 months of age (N=91; *N* girls = 45; *M* age = 627.4 days, *SD* age = 27.7 days).

As part of the first laboratory visit, caregivers were interviewed to obtain demographic indices and family SES data using the modified Kuppuswamy Scale (***Mohd Saleem, 2020***). This scale classifies SES using occupation, education, and family income. *Table 1* shows how high and low SES families (using a median split on the Kuppuswamy SES Score) were distributed along multiple demographic indices.

## Materials

### Visual cognition task

We used a preferential looking change detection task (***Ross-Sheehy et al., 2003***) with a 10 s trial duration (see ***Wijeakumar et al., 2019***; ***Delgado Reyes et al., 2020***). A 42-inch LCD monitor that was connected to a PC running Experiment Builder was used to display the stimuli. Looking data were collected at 500 Hz using an Eyelink 1000 Plus eye-tracker (SR Research). Where eye-tracking data were not available, looking data were collected with a webcam and hand-coded at 30 frames-per-second using ***Datavyu, 2014***.

Infants sat on their mother's lap 100 cm from the screen. A target sticker was placed on the infant's forehead so the eye-tracking system could track head movement. The stimuli consisted of two side-by-side flickering displays composed of colored squares (Figure 1A). One side contained the 'change' display and the other contained the 'no-change' display. Each stimulus display area was 29.5 cm in width and 21 cm in height, with a 21 cm gap between the display on the left and right (each coloured square was approximately 5cm x 5cm). The displays had a solid grey background. The colours of the squares presented on each display were selected from a set of nine colours: green (RGB: 0, 153, 0), brown (128, 64, 32), black (0, 0, 0), violet (128, 0, 128), cyan (128, 255, 255), yellow (255, 255, 0), blue (0, 0, 255), white (255, 255, 255), and red (255, 0, 0). The set size (number of items) was the same between the two displays and remained constant during the 10 s trials. The colors on a display were always different from each other but colors could be repeated between the displays.

### Standardized assessments

We collected standardized psychomotor and cognitive assessments in year 1 using the Mullen Scales of Early Learning (MSEL). In year 2, we used the Ages and Stages Questionnaire (3rd edition; ASQ). We switched to the ASQ in year 2 because preliminary testing with the MSEL revealed many questions that were not relevant to the rural setting in Shivgarh.

### Air quality monitoring

Air quality data were acquired using the Air Visual Node monitor (model B01MF6X1 YK, Atlanta Healthcare, Inc). We selected this device based on validation data comparing the device to a reference Beta Attenuation Monitor (BAM) in Beijing, China during June of 2015 (*Molenar, 2016*). Daily concentration estimates from the two devices correlated highly ($R^2$=0.96). We purchased 20 devices.

## Procedure

### Visual cognition task

Children were tested individually. The task began with a 5-point calibration sequence. Next, the visual cognition task began (see Figure 1A). The squares simultaneously appeared for 500ms and disappeared for 250ms during the 10 s trials. For the 'no-change' display, the colors of the squares remained constant throughout the trial. For the 'change' display, one of the squares changed color after each disappearance. The changing square was randomly selected, and its color was derived from the set of colors not currently present in that display. Consistent with prior studies, memory load was varied between 1, 2, and 3 items on each side for 6- and 9-month-old infants and 2, 4, and 6 items on each side for 18- and 21-month-old infants. Infants were presented with 36 total trials in six blocks of six trials. Each block contained three trials (one for each load) for each change side. Participants could take breaks between blocks. Participants completed on average 20.99 trials in year 1 (SD = 9.72) and 26.25 trials in year 2 (SD = 9.66).

### Standardized assessments

Standardized assessments were conducted in a quiet room with trained staff. The MSEL was administered in Year 1, while the ASQ was administered in Year 2. The ASQ questionnaire for each infant was selected using the online ASQ calculator (https://agesandstages.com/free-resources/asq-calculator/). We adapted the ASQ administration to improve the reliability of the data. Specifically, a trained assessor administered the ASQ in collaboration with the parent. In cases where ASQ questions asked about behaviors that could be elicited (e.g. 'When you ask your child to, does he go into another room to find a familiar toy or object?'), these tasks were completed 'live', ensuring that the child was given ample time. If a question was not amenable to live assessment, the mother's verbal report was taken as the response.

### Use of the Air Visual Node device

During each in-home assessment period, air quality data were collected across 3 days with the monitor placed at roughly head height in the room where the child typically slept or spent most of their time (see Figure 2A). The monitor was inaccessible to children. Air quality was monitored in 3-monthly intervals in-between the Year 1 and Year 2 laboratory visits for a maximum of 6 collection periods for each household. Data were collected from the AVN device every 10 s. Households had to have at least 5 hr of recorded data in any given round for that round to be included in analysis.

Data from the air quality monitors were downloaded after each 3-day period and quality checked. After 8–10 months of use, a few of the devices were returning very high or very low values. Henceforth, devices were regularly cleaned. After cleaning, multiple devices were tested in the same room to ensure they were returning the same air quality readings. Any devices returning erroneous values were retired. By the end of the 2-year study, 11 devices had been retired.

## Methods of analysis

### Visual cognition measures

The eye-tracking data were exported on a frame-by-frame basis using SR research Data Viewer. The area of interest around the two objects was increased to match video coded data which coded looking to the left, right, or away. Where there was no recorded eye-tracking data, the hand-coded video data were used instead. Of the 9956 total trials used to calculate the scores, 3047 trials (30.6%) were manually coded. We re-coded 17% of the data to check reliabilities. Reliabilities were very good with a mean Kappa for the 6-month cohort of 0.73 and a mean Kappa for the 9-month cohort of 0.83.

The looking data were read into R and pre-processed using the R package eyetrackingR (*Dink and Ferguson, 2021*). For the change preference measure, trials where more than 75% of the data

**Table 2.** Baseline model for the change preference scores.

Model parameters for linear mixed effect model assessing the impact of year, load, SES score based on the Kuppuswamy scale, age cohort, and visual dynamics in LookingWindow1 on the 'first-look no-change' change preference scores (baseline change preference model).

| Variable | Estimate | Std. Error | DF | t value | Pr(>\|t\|) |
|---|---|---|---|---|---|
| (Intercept) | 0.488 | 0.054 | 534.793 | 9.062 | <0.001 |
| Year | −0.044 | 0.107 | 742.433 | −0.411 | 0.681 |
| **Load1** | **0.029** | **0.009** | **857.116** | **3.130** | **0.002** |
| Load2 | −0.001 | 0.009 | 854.106 | −0.114 | 0.909 |
| SES | −0.006 | 0.013 | 527.033 | −0.417 | 0.677 |
| LookingWindow1 | −0.067 | 0.065 | 548.916 | −1.038 | 0.300 |
| **Age** | **0.027** | **0.014** | **183.400** | **1.913** | **0.057** |
| Year:SES | 0.036 | 0.027 | 715.570 | 1.350 | 0.177 |
| Year:LookingWindow1 | 0.075 | 0.129 | 731.071 | 0.582 | 0.561 |
| SES:LookingWindow1 | 0.006 | 0.016 | 532.948 | 0.381 | 0.704 |
| Year:SES:LookingWindow1 | −0.054 | 0.033 | 710.951 | −1.665 | 0.096 |

was recorded as not looking at the screen were excluded. Initial analyses of these change preference scores revealed that the change preference measure was not robust longitudinally, that is, year 1 change preference scores did not predict year 2 change preference scores (note: a full correlation table for key measures used in the present study can be found in the supplementary materials; see *Supplementary file 1*). This was the case for the present sample as well as a longitudinal sample from urban UK infants collected as part of a complementary study (see S. Forbes, K. Jackson, K. Mee, J. McCarthy, L. Delgado-Reyes, M. Tiwari, V. Singh, A. Kumar, J. P. Spencer, A longitudinal assessment of the development of visual working memory accross cultures. *manuscript in preparation* for details). Additional analyses from a forthcoming paper (Forbes et al., *manuscript in preparation*) revealed that sorting trials based on where infants are looking at the start of the first 'change' display (i.e. 1000ms) yields two measure that *are* stable longitudinally – a 'first-look change' score and a 'first-look no-change' score. This makes sense as prior to the first change on the screen participants have no way of knowing which is the 'no-change' side and which is the 'change' side. Furthermore, the demands placed on visual cognition differ depending on the starting location. If infants start on the 'no-change' side, they should notice the 'sameness' if the number of items is within their visual working memory capacity and release fixation due to a lack of visual novelty. Thus, 'first-look no-change' trials might be particularly sensitive to individual differences in visual working memory capacity (for discussion, see Forbes et al., *manuscript in preparation*). By contrast, if infants start on the 'change' side, they should

**Table 3.** Model parameters for linear mixed effect model assessing the impact of year, load, SES and age cohort on the shift rate (baseline visual processing speed model).

| Variable | Estimate | Std. Error | DF | t value | Pr(>\|t\|) |
|---|---|---|---|---|---|
| (Intercept) | 0.631 | 0.014 | 196.30 | 45.260 | <0.001 |
| Year | 0.016 | 0.017 | 893.40 | 0.963 | 0.336 |
| **Load1** | **0.040** | **0.011** | **812.80** | **3.485** | **<0.001** |
| Load2 | 0.001 | 0.011 | 811.70 | 0.071 | 0.943 |
| SES | 0.002 | 0.003 | 207.00 | 0.545 | 0.587 |
| Age | 0.012 | 0.028 | 195.80 | 0.429 | 0.668 |
| Year:SES | −0.007 | 0.004 | 928.20 | −1.553 | 0.121 |

**Table 4.** Model parameters for linear models describing effects of age cohort and SES on standardized cognitive scores (baseline standardized cognitive models).
Measures included are MSEL composite T-score and ASQ problem solving score.

| Measure | Variable | Estimate | Std. Error | t value | Pr(>|t|) |
|---|---|---|---|---|---|
| Mullen Composite | (Intercept) | 101.299 | 0.960 | 105.562 | <0.001 |
| | Age | −0.345 | 1.920 | −0.180 | 0.857 |
| | SES | 0.769 | 0.232 | 3.308 | 0.001 |
| ASQ Problem Solving | (Intercept) | 30.581 | 0.955 | 32.032 | <0.001 |
| | Age | −5.549 | 1.909 | −2.907 | 0.004 |
| | SES | 0.923 | 0.231 | 3.998 | <0.001 |

readily notice the novelty and remain fixated on that display. Note that sorting trials based on where infants are looking at a key moment in time is a standard approach in the infant eye-tracking literature (e.g. see *Fernald et al., 1998*).

Infants' looking data were sorted into two types of trials (first-look no-change, first-look change) based on where they were looking at the onset of the first change (1000ms after trial start). Unfortunately, 1030 trials (13.1% of the total trials) had missing data within the time window from 1000 to 1100ms (e.g. due to a failure to track the eye in this window). Thus, we allowed the 'first-look' classification to be determined based on the first frame of non-missing eye-tracking data up to 2500ms (which spanned from the onset of the first change display [1000ms] through two display +delay periods). This allowed us to classify an additional 613 trials, yielding a total trial loss of 417 trials (5.3%) due to a failure to classify the 'first-look' status. The change preference analysis focused on the analysis period from 1750ms to 6750ms. We trimmed the last few seconds of data from each trial as the number of eye-tracking samples diminished as attention waned (see Forbes et al., *manuscript in preparation* for a detailed analysis justifying this time window).

The shift rate measure was taken from the full length of any trial. This measure counted the number of switches participants made from one side of the screen to the other divided by the number of seconds that participants were looking at the display, resulting in shifts per second.

The baseline model for each change preference measure was a linear mixed-effect model with year (1 or 2), working memory load (low, medium or high), SES (Kuppuswamy SES score), and age cohort (6 or 9 months) as independent variables. We also included the mean proportion looking to the first fixated item in the first time window (1–750ms) before any visual changes had been introduced to get an initial measure of visual dynamics for each infant (LookingWindow1). Year and age cohort were difference-coded, SES was centered, and load was input as a factor. To allow for individual differences across year and load, a random intercept for each participant was included. To arrive at a minimal baseline model, we began with a model that only included main effects. We then introduced two-way, three-way, and four-way interactions, only including interactive effects that showed evidence of improving the model fit. For the 'first-look no-change' measure, the final baseline model included

**Table 5.** Model parameters for linear models assessing the baseline variability of SES, cooking fuel, and age cohort on the air quality reading (baseline air quality models).

| Model | Variable | Estimate | Std. Error | t value | Pr(>|t|) |
|---|---|---|---|---|---|
| Cooking Fuel | (Intercept) | 1.461 | 2.767 | 0.528 | 0.598 |
| | Cooking Fuel1 | 10.866 | 4.517 | 2.405 | 0.017 |
| | Cooking Fuel2 | −1.519 | 3.119 | −0.487 | 0.627 |
| | Age | −0.548 | 4.213 | −0.13 | 0.897 |
| SES | (Intercept) | −0.004 | 2.099 | −0.002 | 0.999 |
| | SES | −1.152 | 0.511 | −2.255 | 0.025 |
| | Age | −1.714 | 4.198 | −0.408 | 0.683 |

**Table 6.** Model parameters for linear mixed-effect model assessing the impact of air quality (AQI) on the baseline change preference score model which included Year, Load, SES, Age Cohort, and LookingWindow1 as predictors (see Table 2).

| Variable | Estimate | Std. Error | DF | t value | Pr(>|t|) |
|---|---|---|---|---|---|
| (Intercept) | 0.494 | 0.054 | 535.80 | 9.186 | <0.001 |
| Year | −0.044 | 0.107 | 741.90 | −0.411 | 0.681 |
| Load1 | 0.029 | 0.009 | 858.10 | 3.135 | 0.002 |
| Load2 | −0.001 | 0.009 | 855.20 | −0.124 | 0.901 |
| SES | −0.009 | 0.014 | 524.70 | −0.663 | 0.508 |
| LookingWindow1 | −0.075 | 0.065 | 549.90 | −1.155 | 0.249 |
| Age | 0.026 | 0.014 | 184.70 | 1.810 | 0.072 |
| AQI | 0.000 | 0.000 | 181.80 | −0.654 | 0.514 |
| Year:SES | 0.040 | 0.027 | 720.10 | 1.471 | 0.142 |
| Year:LookingWindow1 | 0.075 | 0.128 | 730.70 | 0.585 | 0.559 |
| SES:LookingWindow1 | 0.010 | 0.016 | 530.80 | 0.607 | 0.544 |
| Year:AQI | 0.001 | 0.000 | 965.10 | 2.431 | 0.015 |
| Year:SES:LookingWindow1 | −0.057 | 0.033 | 714.90 | −1.750 | 0.080 |

all main effects and the following interactions: Year x SES, Year x LookingWindow1, SES x Looking-Window1, and Year x SES x LookingWindow1 (see *Table 2*). For the 'first-look change' measure, the final baseline model included all main effects (see Table 10).

All models were assessed for fit based on a Q-Q plot of the residuals and the R package DHARMa (*Hartig, 2020*). Analyses reported in the tables use the *p* values calculated using Satterthwaite's method from the package lmerTest (*Kuznetsova et al., 2017*) to enable the reader to glean the direction of all effects. In addition, the contribution of each significant effect to a model was assessed using type 3 Wald Chi-squared tests, with effect sizes reported in text using the effectsize package in R (*Ben-Shachar et al., 2020*). In all models we aimed to theoretically motivate the inclusion of all effects and limit spurious correlations by only adding effects where they were thought to contribute a priori. In the case of interactions, models were tested to see whether interactions improved model fit through formal model comparison, and interactions were not included if they did not contribute to model fit.

The baseline model for shift rate was arrived at using a similar procedure. The final baseline model included all main effects and a Year x SES interaction (see *Table 3*).

**Table 7.** Model parameters for linear mixed-effect model assessing the impact of air quality (AQI) on the baseline visual processing speed (shift rate) model which included Year, Load, SES, and Age Cohort as predictors (see Table 3).

| Variable | Estimate | Std. Error | DF | t value | Pr(>|t|) |
|---|---|---|---|---|---|
| (Intercept) | 0.631 | 0.014 | 195.40 | 45.655 | <0.001 |
| Year | 0.016 | 0.017 | 894.50 | 0.977 | 0.329 |
| Load1 | 0.040 | 0.011 | 813.20 | 3.501 | <0.001 |
| Load2 | 0.001 | 0.011 | 812.10 | 0.045 | 0.964 |
| SES | 0.001 | 0.003 | 205.10 | 0.225 | 0.823 |
| Age | 0.009 | 0.028 | 195.30 | 0.318 | 0.750 |
| AQI | −0.001 | 0.000 | 192.10 | −2.166 | 0.032 |
| Year:SES | −0.006 | 0.004 | 928.60 | −1.493 | 0.136 |

**Table 8.** Model parameters for linear model assessing the impact of air quality (AQI) on the baseline Mullen model (Composite T-Score) which included Age Cohort and SES as predictors (see Table 4).

| Variable | Estimate | Std. Error | t value | Pr(>\|t\|) |
|---|---|---|---|---|
| (Intercept) | 101.301 | 0.961 | 105.398 | <0.001 |
| Age | −0.377 | 1.923 | −0.196 | 0.845 |
| **SES** | **0.746** | **0.236** | **3.163** | **0.002** |
| AQI | −0.019 | 0.032 | −0.596 | 0.552 |

**Table 9.** Model parameters for linear models assessing the impact of air quality (AQI) on the baseline ASQ model which included Age Cohort and SES as predictors (see Table 4).
For comparison with prior work, we include analyses of the ASQ Problem Solving score as well as Fine and Gross Motor scores.

| Measure | Variable | Estimate | Std. Error | t value | Pr(>\|t\|) |
|---|---|---|---|---|---|
| ASQ Problem Solving | (Intercept) | 30.563 | 0.955 | 32.004 | <0.001 |
| | **Age** | **−5.617** | **1.910** | **−2.941** | **0.004** |
| | **SES** | **0.890** | **0.233** | **3.817** | **<0.001** |
| | AQI | −0.030 | 0.031 | −0.983 | 0.327 |
| ASQ Fine Motor | (Intercept) | 34.224 | 1.039 | 32.944 | <0.001 |
| | **Age** | **−5.655** | **2.078** | **−2.721** | **0.007** |
| | **SES** | **0.815** | **0.254** | **3.214** | **0.002** |
| | AQI | −0.025 | 0.033 | −0.748 | 0.455 |
| ASQ Gross Motor | (Intercept) | 36.849 | 1.061 | 34.733 | <0.001 |
| | **Age** | **−8.221** | **2.122** | **−3.874** | **<0.001** |
| | **SES** | **0.571** | **0.259** | **2.206** | **0.029** |
| | AQI | −0.054 | 0.034 | −1.582 | 0.115 |

**Table 10.** Model parameters from the baseline mixed-effects model assessing the effects of Year, Load, SES, LookingWindow1 and Age cohort on the first look change measure.

| Variable | Estimate | Std. Error | DF | t value | Pr(>\|t\|) |
|---|---|---|---|---|---|
| (Intercept) | 0.592 | 0.054 | 613.50 | 11.009 | <0.001 |
| **Year** | **−0.030** | **0.013** | **966.10** | **−2.353** | **0.019** |
| Load1 | 0.000 | 0.009 | 846.40 | 0.024 | 0.981 |
| Load2 | 0.002 | 0.009 | 847.40 | 0.175 | 0.861 |
| SES | 0.002 | 0.002 | 205.30 | 0.874 | 0.383 |
| LookingWindow1 | 0.057 | 0.065 | 628.30 | 0.882 | 0.378 |
| Age | −0.019 | 0.015 | 188.80 | −1.331 | 0.185 |

## Standardized assessments

We used non-standardized measures from the MSEL and ASQ as there are no standardized scores for rural India. From the MSEL, we use the composite standard T-score as a general measure of cognition. The baseline model for this measure was a linear model with age cohort and SES included as main effects (see *Table 4*). For the ASQ, we used the problem-solving measure. The baseline model was a linear model with the ASQ measure as the dependent variable and age cohort and SES as independent variables (see *Table 4*).

## Data filtering and analysis of air quality data

The air quality data were collated and read into R. Data were inspected to understand the cyclical patterns across days and months. The air quality data were filtered to remove data from any defective devices (see above), and then down-sampled so that we had one observation per hour. This smoothed the data and removed temporal autocorrelation. The down-sampled data were then fit with a generalized additive mixed model (GAMM) in R using the 'bam' function from the package mgcv (**Wood, 2011**). The model included air quality as the dependent variable and a tensor smooth of time of day (in hours) and date, with a cubic regression spline of 11 dimensions for each as independent variables. The model also allowed a smoothed random effect of participant. To reduce autocorrelation, the model was fit with a Rho of 0.6. Temporal autocorrelation was assessed using the itsadug package in R (**van Rij et al., 2020**). The model provided an excellent fit, capturing annual fluctuations by date (see Figure 2B) and daily fluctuations by hour (see Figure 2C).

The participant-level random intercept was extracted from the model, to act as a single data measurement to represent air quality in each home. This measurement was highly correlated with the mean air quality score for each participant, $t(213) = 21.348$, p<0.001. This individual level score was centered around 0, where a positive score indicated a higher air quality score (i.e. poorer air quality) on average than the grand mean (i.e. 185.824). For visualization purposes, Figures 2 and 3 were adjusted by adding the grand mean to the individual score. Note that the participant-level random intercept provides a measurement that is more robust than the mean air quality value as time of day and seasonal effects are included in the model. This was particularly evident in cases where a participant contributed only a small number of samples (in which case, the sample values can be biased due to time of day).

We modelled individual estimates of air quality using a linear model with the AQI random intercept value as the dependent variable and age cohort and cooking fuel type as independent variables (see **Table 5**). A second model for air quality was run but using the SES score (centered) as a replacement for cooking fuel (also in **Table 5**). These models were comparable, showing the strong relationship between cooking fuel and SES (see also, **Table 1**).

## Analysis of air quality associations with cognition

To assess the association of air quality with visual cognition, we added air quality to the baseline model for each dependent variable, that is, for the 'first-look no-change' change preference score (**Table 6**), shift rate (**Table 7**), and the 'first-look change' change preference score (Table 11). An Air Quality x Year interaction was also included in the 'first-look no-change' change preference model as this improved the model fit, assessed using the anova function in R, $X^2(1)=4.824$, p=0.028, $\eta^2$ (partial)=0.005.

To assess the association of air quality with standardized cognitive scores, we added air quality to the baseline model for the MSEL scores (**Table 8**) and the ASQ problem-solving scores (**Table 9**). In

**Table 11.** Model parameters assessing the impact of air quality (AQI) on the baseline mixed-effects model assessing the effects of Year, Load, SES, LookingWindow1 and Age cohort on the first look change measure.

| Variable | Estimate | Std. Error | DF | t value | Pr(>\|t\|) |
|---|---|---|---|---|---|
| (Intercept) | 0.589 | 0.054 | 612.30 | 10.944 | <0.001 |
| Year | −0.030 | 0.013 | 967.10 | −2.395 | 0.017 |
| Load1 | 0.000 | 0.009 | 847.40 | −0.008 | 0.994 |
| Load2 | 0.002 | 0.009 | 848.30 | 0.182 | 0.855 |
| SES | 0.002 | 0.002 | 205.30 | 1.105 | 0.270 |
| LookingWindow1 | 0.062 | 0.065 | 626.90 | 0.952 | 0.341 |
| Age | −0.018 | 0.014 | 189.50 | −1.275 | 0.204 |
| AQI | 0.000 | 0.000 | 185.80 | 1.263 | 0.208 |
| Year:AQI | −0.001 | 0.000 | 968.20 | −1.402 | 0.161 |

addition, following findings from *Guxens et al., 2014*, we conducted analyses on the ASQ fine and gross motor scores to examine whether these measures were associated with air quality (see *Table 9*).

## Results

We focused on three primary measures from the preferential looking task: (1) the shift rate, that is, the number of looks back and forth (per second), (2) the change preference score for trials where infants' 'first look' was to the 'no change' side, and (3) the change preference score for trials where infants' 'first look' was to the 'change' side. Use of the shift rate measure was motivated by prior work showing that visual processing speed in infancy is predictive of longer-term cognitive outcomes (*Rose et al., 2012*). The latter two measures are an adaptation of the standard 'change preference' looking measure from this task. Note that because the 'first-look no-change' measure starts at 0 (i.e. looking to no change) and the 'first-look change' measure starts at 1 (i.e. looking to change), chance performance is no longer anchored at 0.5. Although this differs from prior work, the change was motivated by evidence that the standard measure is not predictive longitudinally, while the adapted measures are (see Methods for discussion). Note further that we focus on the 'first look no-change' measure below (referred to as the 'change preference' score for simplicity), as the only significant finding for

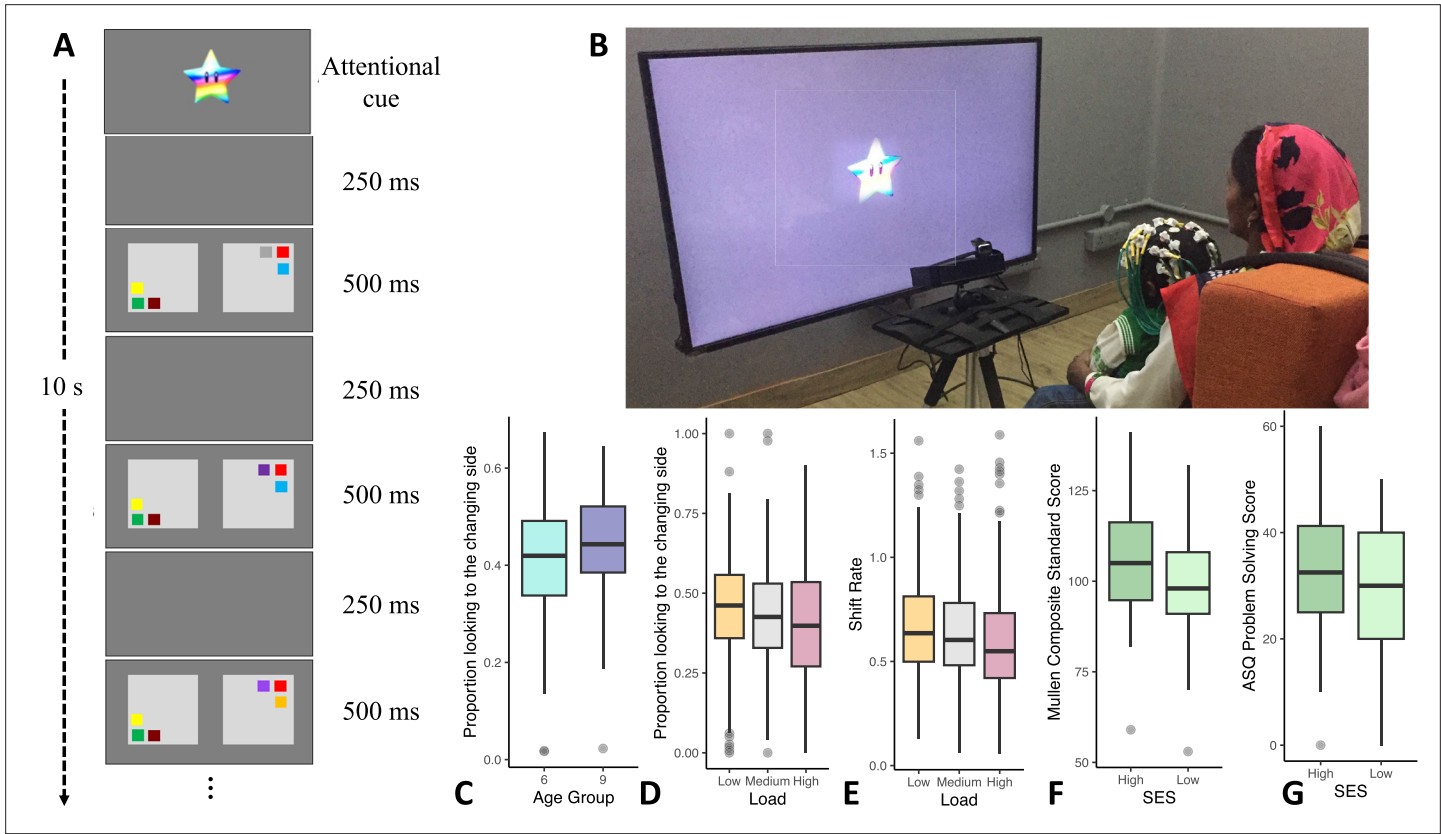

**Figure 1.** Variations in infants' cognitive performance. (**A**) A schematic of the visual cognition task. (**B**) An infant performing the task. (**C**) The 6-month-old cohort (N = 107) had lower 'first-look no-change' change preference scores relative to the 9-month-old cohort (N = 106). (**D**) Infants showed higher change preference scores in the low memory load condition (N = 210) relative to the medium (N = 208) and high loads (N = 209). (**E**) Infants had faster visual processing speed (higher shift rates) in the low load condition (N = 206) relative to the medium (N = 206) and high loads (N = 205). (**F**) Standardized composite scores from the Mullen Scales of Early Learning (MSEL) in year 1 were higher for high SES infants (N = 97) than for low SES infants (N = 112). (**G**) Problem-solving scores from the Ages and Stages Questionnaire (ASQ) in year 2 were higher for high SES infants (N = 84) than for low SES infants (N = 96). Note that for F and G, a continuous SES score based on the Kuppuswamy Scale (see *Mohd Saleem, 2020*) was used in analysis, but this was median-split for ease of visualization. Line in boxplots shows the median, lower and upper hinges show the first and third quartiles, lower and upper whiskers extend to the smallest and largest point no more than 1.5 * the interquartile range from the closest hinge respectively, and data beyond teh whiskers are outlying and are plotted individually.

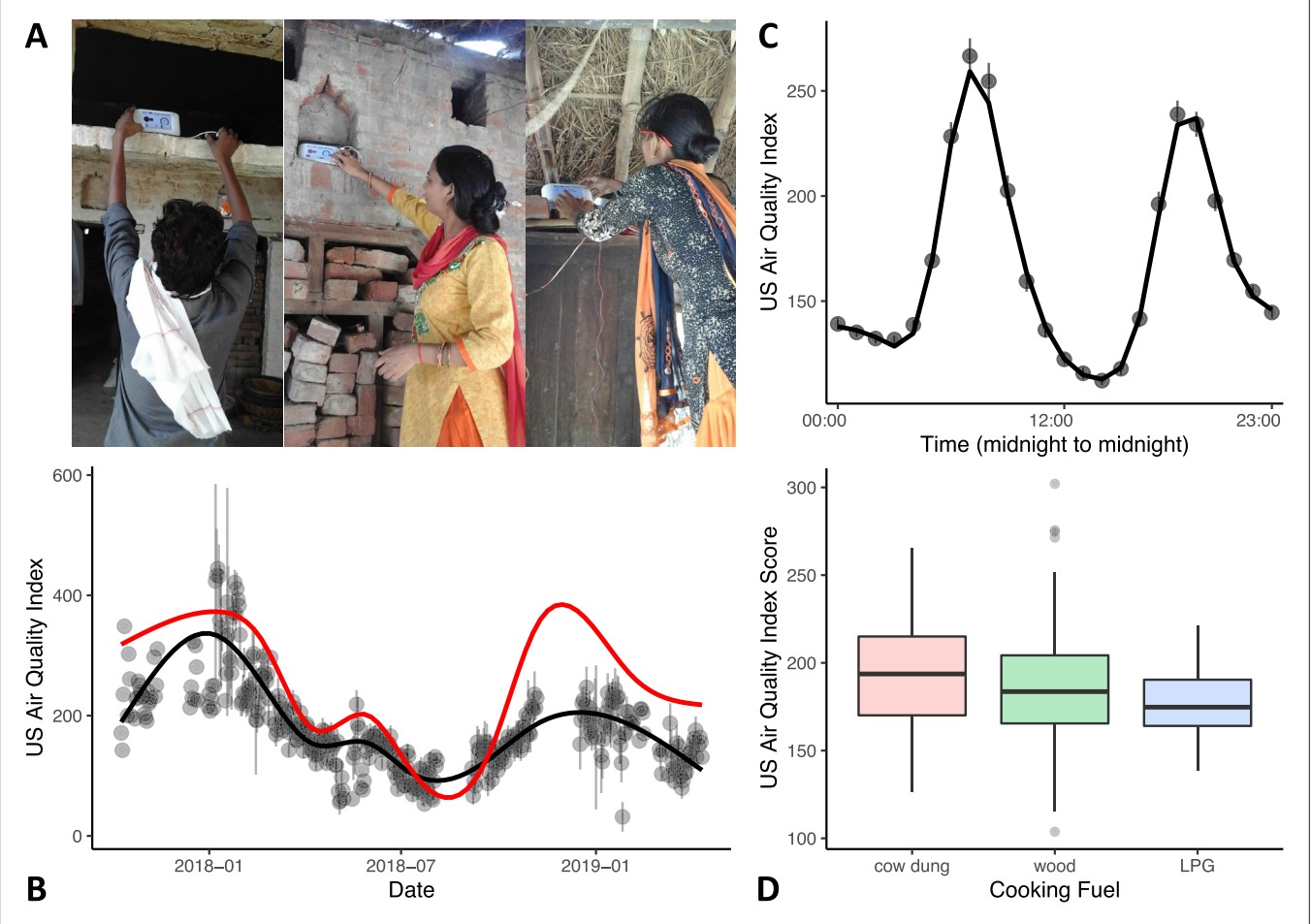

**Figure 2.** Variations in in-home air quality (PM$_{2.5}$) by year, by day, and by type of cooking fuel. (**A**) Three examples of in-home sensor placement for households of varying SES levels. (**B**) Variations in in-home air quality over years in the study (participants contributing data = 215). Black dots show mean air quality index over each 3-day assessment period with standard errors indicating variability over households collected on the same day. Black line shows our model fit through these data. Red line shows best-fitting curve from outdoor air quality observations recorded in Lucknow, India. (**C**) Daily variations in in-home air quality with peaks at meal preparation times (participants contributing data = 215). Points indicate raw data (with standard errors), the line indicates our model fit. (**D**) Plots showing poorer in-home air quality for households that used cow dung for cooking fuel (N = 25) relative to wood (N = 152) and liquified petroleum gas (LPG; N = 38). Boxplot details are same as in *Figure 1*.

the 'first look change' measure was a decrease in the change preference score from year 1 to year 2 (see *Table 10* and *Table 11*).

Six-month-old infants showed lower change preference scores than 9-month-old infants (*Figure 1C*), $X^2(1)=3.66$, p=0.056 $\eta_p^2 = 0.02$ (for full results, see *Table 2*). In addition, change preference scores decreased as the memory load increased, $X^2(2)=12.53$, p=0.002 $\eta_p^2 = 0.01$ (see *Figure 1D*). Similar effects were observed in an analysis of infants' rate of looking back and forth between the displays (i.e. shift rate; see *Table 3*). Infants' shift rate decreased as the memory load increased, $X^2(2)=16.52$, p<0.001 $\eta_p^2 = 0.02$ (see *Figure 1E*). In addition, infants' standardized cognitive scores in year 1, $F(1)$ = 10.94, p=0.001 $\eta_p^2 = 0.050$, and year 2, $F(1)$ = 15.99, p<0.001 $\eta_p^2 = 0.080$, were consistently lower for low SES infants (see *Figure 1F and G* and *Table 4*).

We recorded in-home air quality using a laser particle sensor (Air Visual Node, Atlanta Healthcare, Inc) placed in the home (see *Figure 2A*) for 3 continuous days during each assessment period. Field workers were instructed to place the sensor in the room where infants slept or spent most of their time. We re-assessed air quality up to six times for each family (every three months in-between lab visits; *M* visits = 4, SD = 1.16). These data were modelled and a participant-level score which adjusted for season and time of day was extracted for use in the analyses (see Methods). We focused on PM$_{2.5}$ concentrations expressed as US Air Quality Index (AQI) values. This index maps PM$_{2.5}$ concentrations

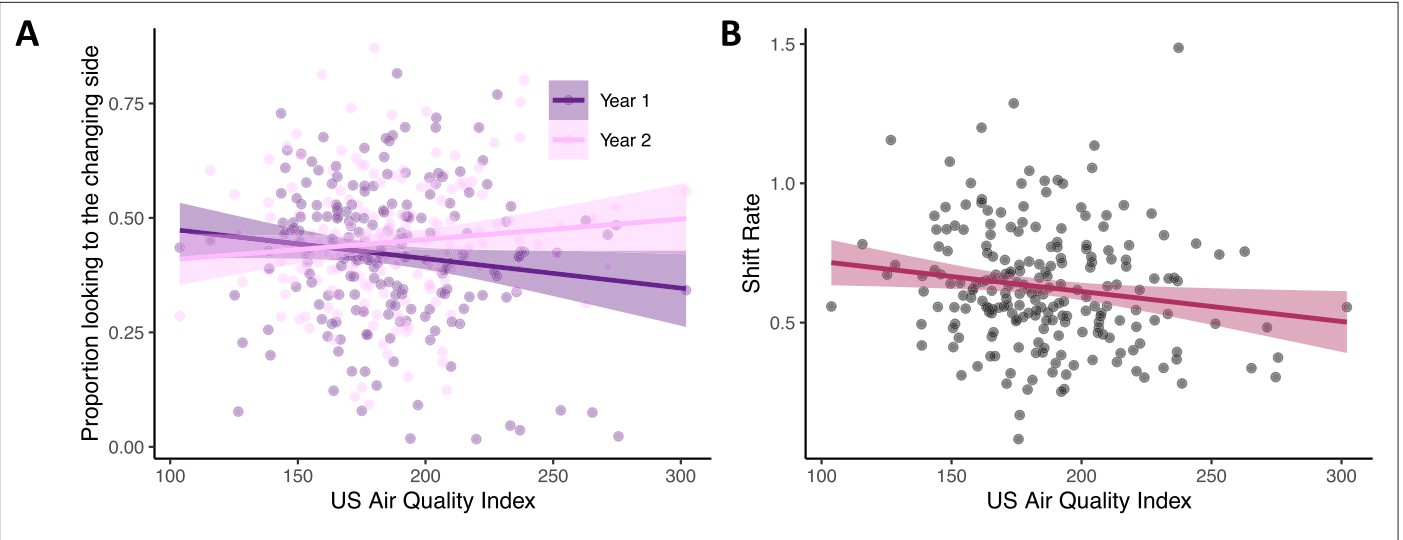

**Figure 3.** Poor air quality is associated with impaired visual cognition in infancy. (**A**) Infants from households with better air quality (lower AQI scores) had higher visual working memory scores in year 1 (see dark purple line; N = 199) relative to effects in year 2 (pink line; N = 179). (**B**) Infants from households with better air quality (lower AQI scores) also had faster visual processing speeds (higher shift rates; N = 213). Dots in both panels show raw data, line indicates linear trend with the ribbon indicating the 95% confidence interval.

measured in µg/m$^3$ to a more intuitive categorical scale where AQI values under 50 are good, values from 51 to 100 indicate moderate air quality, values 101–150 indicate air that is unhealthy for sensitive groups, and values higher than 151 are considered unhealthy extending up to hazardous (>301). Note that AQI values can be readily converted to µg/m$^3$ using on-line calculators (e.g. https://www.airnow. gov/aqi/aqi-calculator/).

As can be seen in *Figure 2B*, air quality in Shivgarh was quite poor, with AQI values often higher than 151, with an overall mean of 207. This is comparable to recent AQI data for the northern states in India that ranged from 186 to 267 (*Pandey et al., 2021*). We also compared in-home air quality to daily outdoor air quality from the nearest monitoring station in Lucknow (red line in *Figure 2B*; see Central Pollution Control Board in India: https://cpcb.nic.in). Annual fluctuations in in-home air quality generally mirrored fluctuations in outdoor air quality in Lucknow, although with a lower peak in the winter of 2018–2019 (for similar seasonal variations in indoor and outdoor air quality, see *Rohra and Taneja, 2016*). We note that this lower peak occurred during a pause in our indoor air quality data collection; thus, we cannot evaluate if this is a mismatch or simply a result of limited data during this period in the indoor air quality model.

In-home air quality (PM$_{2.5}$) was strongly influenced by daily meal preparation, with peaks in poor air quality occurring during meal preparation times (see *Figure 2C*; see *Mukhopadhyay et al., 2012* for similar meal preparation findings). Air quality was poorest in households using cow dung for fuel and best in households using LPG, $F_{(2)} = 3.23$, p=0.042 $\eta_p^2 = 0.030$ (see *Figure 2D*; *Table 5*). Note that we were not able to look at the impact of cooking fuel independently from SES as most families that used LPG fuel fell into the high SES tertile (see *Table 1*).

Next, we explored the association between air quality (PM$_{2.5}$) and cognitive measures while controlling for both age cohort and SES (see Methods). Critically, infants living in homes with **poor air quality had poorer visual cognitive performance**. Infants from households with poorer air quality had lower change preference scores in year 1, an effect that was attenuated in year 2, $X^2(1)=5.91$, p=0.015 $\eta_p^2 = 0.006$ (see *Figure 3A* and Year x Air Quality interaction in *Table 6*). We also found a strong negative association of air quality with visual processing speed (shift rate), $X^2(1)=4.69$, p=0.03 $\eta_p^2 = 0.02$ (see Air Quality main effect in *Table 7*). As can be seen in *Figure 3B*, infants from households with poor air quality showed slower rates of visual processing. Note that both effects shown in *Figure 3* were robust in models that controlled for SES using the modified Kuppuswamy Scale which aggregates effects of family occupation, education, and income (see *Mohd Saleem, 2020*). Moreover, in both cases, models that included air quality captured a significant proportion of variance above and beyond the baseline cognitive models: models comparing the baseline change preference model

(*Table 2*) to a model that included air quality (*Table 6*) showed that the air quality model captured a greater proportion of variance in the change preference scores, $X^2(2)=6.35$, p=0.04; similarly, comparing the baseline visual processing speed model (*Table 3*) to a model that included air quality (*Table 7*) showed that the air quality model captured a greater proportion of variance in processing speed, $X^2(1)=4.74$, p=0.03.

One concern with these findings is that change preference and shift rate are measured in the same task and are correlated. Thus, it is possible the findings above reflect some shared variance rather than separate effects. To examine this, we re-ran the change preference / air quality analysis, adding shift rate as a predictor of the change preference score. This analysis revealed a main effect of shift rate, $X^2(1)=17.36$, p<0.001 $\eta_p^2 = 0.02$, confirming the relationship with the change preference score; however, the Load and Year x Air Quality interaction remained significant (see *Supplementary file 2*). Thus, the effects shown in *Figure 3A* are statistically robust, even when shift rate is included in the change preference model. We also re-ran the shift rate / air quality model, including the change preference score as a predictor. This analysis revealed a main effect of change preference score, $X^2(1)=23.02$, p<0.001 $\eta_p^2 = 0.02$. Critically, however, the Load and Air Quality main effects remained significant even when the change preference score was included as a predictor (see *Supplementary file 3*).

Another key question is why the change preference measure only showed a robust association with air quality in the first year. As discussed in the Methods, we modulated the working memory load in year 2 to make the task more challenging and age-appropriate; thus, it is possible we made the task too hard for some infants, dampening our ability to detect individual differences in working memory. To explore this possibility, we re-ran the change preference / air quality analysis, only using data from load 2 trials (i.e. the 'medium' load in year 1 and the 'low' load in year 2; see Methods). In this analysis with identical task stimuli, we still found a significant Year x Air Quality interaction, $F(1,340) = 6.83$, p=0.009 $\eta_p^2 = 0.02$ (see *Supplementary file 4*).

Finally, we note that air quality did not significantly impact standardized cognitive scores in year 1 (measured using the Mullen Scales) or year 2 (measured using the ASQ; see *Tables 8 and 9*). This may indicate that the effects of air quality are specific to the visual cognitive system rather than impacting the more general aspects of cognition and psychomotor function assessed by these measures.

## Discussion

The present findings indicate that poor air quality ($PM_{2.5}$) is associated with slower visual processing speed in the first two years of life and poorer visual working memory scores in year 1. These negative impacts were evident only for looking-based measures of cognition. Our results contrast with findings from prior studies that have failed to show an association between outdoor air quality and cognition in early development. It is possible that this difference reflects the broader range of $PM_{2.5}$ exposure in our sample. For instance, Guxens and colleagues (*Guxens et al., 2014*) reported no systematic relationships between air quality and cognitive measures across six European birth cohorts (although they did find effects on psychomotor function); however, air quality ranged from AQI values of 53–72. Our mean AQI value (207) was three to four times higher. Thus, infants in the present report were exposed to much poorer air quality which might explain the strong relationship with visual cognition.

It is also possible that *indoor* levels of $PM_{2.5}$ are critical to cognition in infancy. Most prior studies have looked at outdoor air quality, although several studies have reported a negative impact of poor indoor air quality on cognition as assessed via questionnaires (*Midouhas et al., 2019*; *Gonzalez-Casanova et al., 2018*; *Vrijheid et al., 2012*; *Midouhas et al., 2018*). Interestingly, *Vrijheid et al., 2012* found that use of a gas cooker in the home showed a stronger negative association with standardized cognitive scores *after* 14 months. Our findings extend this work by also showing an association between poor indoor air quality and visual cognition as early as 6 months of age.

One question raised by our findings is why visual working memory effects were isolated to year 1. We investigated one possibility – that the working memory task was too difficult in year 2. Results suggested that this was not the case: when we compared identical conditions across years 1 and 2 we still found an inverse relationship between change preference scores and air quality in year 1 but not in year 2. Another possibility is that the impact of indoor air quality on visual cognition wanes in year 2. Our findings for shift rate argue against this possibility as these findings were robust across both years. The robustness of the shift rate effect may suggest that visual processing speed is a particularly

sensitive measure in infancy, consistent with other work showing that visual processing speed in infancy is predictive of schooling outcomes 11 years later (*Rose et al., 2012*). A third possibility for why the impact of air quality on change preference scores wanes in year 2 is that other factors which are not correlated with air quality have a stronger impact on visual working memory in year 2. For instance, we are currently examining how interactions with caregivers impact infants' visual working memory abilities. We suspect these interactions have a strong influence on visual working memory development, particularly in year 2 after extended interactions accumulate over time. Interestingly, infant-led interactions – which have been shown to support working memory development (*Landry et al., 2006*; *Perone and Spencer, 2013*) – appear to be quite frequent in low SES families in our sample. It is possible such positive influences counter the impact of poor air quality in some families.

Another interesting result from the present study was the specificity of our findings to the visual cognition task. In particular, although results from the Mullen and Ages and Stages Questionnaire showed robust relationships with SES suggesting good sensitivity, these measures showed no associations with air quality. Thus, air quality may have specific effects in infancy, targeting early emerging cognitive systems. Future work will be needed to determine if this is the case. For instance, it would be interesting to examine if air quality has an impact on auditory and/or statistical learning processes in the first two years of life (*Saffran et al., 1996*).

Strengths of the current study include our use of multiple measures to assess infant cognition including looking-based measures, the longitudinal design, inclusion of a large sample, and a high density of air quality measurements for each household. We also included a continuous range of SES families from the same micro-cultural context, allowing us to tease apart influences of SES while holding culture relatively constant.

Regarding limitations, we note that effect sizes were relatively low in the present study. In addition, there was some instability in the air quality devices over time (see Methods). Note that although laser particle sensors have shown robust correlations with beta attenuation monitors in laboratory and field tests (*Zhang and Srinivasan, 2020*), it would be ideal in future work to use personal air quality monitors such as gravimetric devices that correct light scatter and use weighted $PM_{2.5}$. This would enable a direct, localized measure of the air each child is exposed to.

Another limitation of the present study was the absence of more detailed information about cooking practices in the home. Although we identified the primary cooking fuel used in each household, it is likely some households used multiple cooking methods, and other fuels for warming, smoking, and so on (*Rohra and Taneja, 2016*; *Mukhopadhyay et al., 2012*). Future work will also be needed to carefully examine whether poor air quality has a causal influence on cognition in early development. Given recent data on the impact of poor air quality on the developing brain in animals, future work using neuroimaging tools might be particularly useful to clarify mechanistic pathways in infancy.

Our data suggest that global efforts to improve air quality could have benefits to infants' emerging cognitive abilities. This, in turn, could have a cascade of positive impacts, including positive impacts on families as well as economic consequences as improved cognition can lead to improved economic productivity longer-term and reduce the burden on healthcare and mental health systems (*Moffitt et al., 2011*). Our results showing links between indoor air quality and cooking materials also suggest that efforts to reduce cooking emissions in homes should be a key target for intervention. This requires both increased availability of clean technologies and uptake of such technologies in rural households where traditional methods of meal preparation might be a barrier (see *Mukhopadhyay et al., 2012*). Our findings can motivate both policymakers and families to improve air quality as this should positively boost the neurocognitive health of young infants.

## Acknowledgements

We are grateful to the community of Shivgarh, especially the mothers and babies who participated. We thank the 1000 Dreams Team at CEL – Tarawati Chachi, Hariprasad, Matadeen, Ram Kishore, Mukesh, Arjun, Ashok Kumar Pandey, Shivendra, Raj Kishore, Sunil Shukla, Satya Prakash Singh, Vineeta Maurya, Babita Singh, Saroj, Ritu Singh, Pooja Verma, Beenu Pandey, Jitendra Patel, Amit Patel, Dileep Verma, Saurabh Verma, Ankit Bajpayee, Jyoti Singh, Jay Prakash, Shreya Singhal, Dr. Ravi Keshava Lele, Rajesh Kumar, and Abhishek Singh who was the Operations Lead for the study. We also thank Ranjit Kumar for his administrative and logistical support, and Mohammad Amir, Sudhanshu Srivastav, Deepak Sahu, Vishwadeep Mukherjee, and Amit Tandon for their technological support. Our

gratitude to Dr. Vishwajeet Kumar and Mr. Rakesh Pratap for their support and guidance. Outdoor air quality data from Lucknow were downloaded from https://www.kaggle.com/rohanrao/air-quality-data-in-india. We thank Dan Pope, Elisa Puzzolo, and Prerna Aneja for useful discussions. Bill & Melinda Gates Foundation grant OPP1164153 (JPS).

## Additional information

### Competing interests

Sean Deoni: Sean Deoni has grants or contracts, received consulting fees from and has patents planned, issued or pending with Nestle Nutrition and has received Payment or honoraria for lectures, presentations, speakers bureaus, manuscript writing or educational events from Wyeth Nutrition and Mead Johnson Nutrition. The author has no other competing interests to declare. The other authors declare that no competing interests exist.

### Funding

| Funder | Grant reference number | Author |
| --- | --- | --- |
| Bill and Melinda Gates Foundation | OPP1164153 | John P Spencer |

The funders had no role in study design, data collection and interpretation, or the decision to submit the work for publication.

### Author contributions

John P Spencer, Conceptualization, Resources, Data curation, Software, Formal analysis, Supervision, Funding acquisition, Investigation, Visualization, Methodology, Writing – original draft, Project administration, Writing – review and editing; Samuel H Forbes, Resources, Data curation, Software, Formal analysis, Visualization, Writing – review and editing; Sophie Naylor, Kiara Jackson, Data curation, Formal analysis; Vinay P Singh, Resources, Data curation, Supervision, Methodology, Project administration; Sean Deoni, Conceptualization, Data curation, Supervision, Funding acquisition, Methodology, Project administration, Writing – review and editing; Madhuri Tiwari, Data curation, Supervision, Methodology, Writing – review and editing; Aarti Kumar, Conceptualization, Data curation, Formal analysis, Supervision, Funding acquisition, Methodology, Project administration, Writing – review and editing

### Author ORCIDs

John P Spencer http://orcid.org/0000-0002-7320-144X
Samuel H Forbes http://orcid.org/0000-0003-1022-4676
Vinay P Singh http://orcid.org/0000-0003-1421-8775

### Ethics

The study was approved by the Community Empowerment Lab Institutional Ethics Committee (Ref. No: CEL/2018005). Participants' caregivers provided written informed consent; where caregivers were illiterate, a witness gave signed consent accompanied by a thumb impression of the caregiver in place of a signature.

### Decision letter and Author response

Decision letter https://doi.org/10.7554/eLife.83876.sa1
Author response https://doi.org/10.7554/eLife.83876.sa2

## Additional files

### Supplementary files

• Supplementary file 1. Correlation table showing pairwise correlations for the key measures from the present study. AQI = air quality index; SES = SES score from the Kuppuswamy scale; PropC = 'first-look change' change preference score; PropNC = 'first-look no-change' change preference

score; SR = shift rate. First index number indicates load (1=Low, 2=Medium, 3=High) and second index number indicates year (1 or 2). Colors reflect the strength of the correlation (see bar).

• Supplementary file 2. Model assessing impact of air quality on change preference scores controlling for shift rate. Model parameters for linear mixed-effect model assessing the impact of air quality (AQI) on the baseline change preference score model which included Year, Load, SES, Age Cohort, and LookingWindow1 as predictors, controlling for the shift rate (see *Table 6*).

• Supplementary file 3. Model assessing the impact of air quality on shift rate controlling for change preference scores. Model parameters for linear mixed-effect model assessing the impact of air quality (AQI) on the baseline visual processing speed (shift rate) model which included Year, Load, SES, and Age Cohort as predictors, controlling for the change preference score (see *Table 7*).

• Supplementary file 4. Model parameters from linear model examining the effects of air quality on a change preference model in set size 2 only, across both years. Parameters include year, SES, Looking window 1 and age cohort.

• Supplementary file 5. Full set of assessments carried out in Project INDIA (Infant Neural and Dyadic Interaction Assessment).

• MDAR checklist

### Data availability

All data and code are available at https://osf.io/fspzb/.

The following dataset was generated:

| Author(s) | Year | Dataset title | Dataset URL | Database and Identifier |
|---|---|---|---|---|
| Spencer JP, Forbes S | 2023 | Air quality and visual working memory | https://osf.io/fspzb/ | Open Science Framework, fspzb |

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
