## [Editor Report]

This study presents an important finding on the negative association of indoor air quality with visual cognition in the first two years of life. Key strengths include the longitudinal design, fine-grained measures of indoor air quality, and multi-modal assessment of cognitive functioning in a large sample of infants from families across diverse SES strata. The evidence provided is solid and will be of interest to researchers working in the fields of neurocognition, child development, environmental and public health.

---

## [Decision Letter]

**Decision letter after peer review:**

Thank you for submitting your article "Poor air quality is associated with impaired visual cognition in the first two years of life: a longitudinal investigation" for consideration by *eLife*. Your article has been reviewed by 3 peer reviewers, one of whom is a member of our Board of Reviewing Editors, and the evaluation has been overseen by Chris Baker as the Senior Editor. The following individual involved in the review of your submission has agreed to reveal their identity: Zsuzsa Kaldy (Reviewer #2).

Essential revisions:

1) Please provide more information about the following general aspects of the study methodology and variables examined:

a. A rationale for the recruitment strategy and sample selection (e.g. why the study included two cohorts of infants, aged specifically 6 and 9 months of age);

b. Adding a table or correlation plot (e.g. in supplementary materials) showing how the study variables relate to each other and over time;

c. Clarifying the strategy for model building and covariate adjustment, as certain covariates (e.g. gender) seem to be included in models unevenly ;

d. Commenting on the power to test 2-, 3- and 4-way interactions given the available sample size;

e. Expanding on findings regarding within-subject comparisons.

2) The treatment and analysis of data from the computerized tasks need further clarification, particularly in relation to the following points:

a. Can the authors clarify why they focused only on 'first-look same trials' (i.e. when infants focused on the unchanging stimulus) in their change preference analyses? As suggested by the reviewers, it would be important to provide a comparison by reporting looking time when infants had their first look at the changing stimulus;

b. Analyses were limited to a specific temporal window, and it is unclear how this choice might have influenced the results. Can the authors provide more information about this, for example using the 'time course' plot suggested by Reviewer 3;

c. The authors should comment on potential reasons why change preference scores in this study were on average lower than 0.5 (i.e. chance level) and how this might impact the interpretation of findings;

d. The rationale for using the shift rate measure needs to be expanded given the potential for inter-dependence with change preference score.

3) Re-evaluating and expanding the interpretation of findings:

a. The authors mention that the lack of significant associations between indoor air pollution and questionnaire-based measures of cognition may be due to the measures having lower validity for this type of sample. However, as pointed out by several reviewers, these measures do associate with other study variables, such as SES, as expected. Can the authors revise their conclusions accordingly and expand on alternative potential reasons for the lack of associations with indoor air pollution?

b. Conclusions about associations between indoor air pollution and performance on the visual cognition task should be qualified by stating more clearly that observed effect sizes were generally small, the relationship was not consistent over time and that overall performance on change preference was lower than expected (i.e. on average below 0.5). As suggested by Reviewer 2, the authors should also perform a post hoc analysis to test whether an association between air quality and change preference scores in Year 2 is observed when restricting data to set sizes 2 and 4 only. This will help readers evaluate whether the lack of associations in Year 2 may be due to the task being too difficult or whether other factors may be at play.

*Reviewer #1 (Recommendations for the authors):*

In this study, Spencer et al. examine whether indoor air quality is associated with multiple aspects of infant cognitive functioning, based on longitudinal data from a sample of families from rural India. Briefly, infants were enrolled at either 6 or 9 months of age. Laboratory assessments were performed at baseline and repeated one year later, whereas home visits to measure air quality took place after each laboratory assessment (i.e., baseline and follow-up), and were repeated every three months for a period of one year. The main effects of air quality as well as interactions with factors such as age, gender, and socio-economic status were tested. Based on these data, the authors report an association between poorer air quality and worse performance on a computerized task of visual cognition. Specifically, associations with processing speed were observed at both time points, while associations with working memory were only observed at baseline (year 1). No associations were identified with standardized questionnaire-based measures of psychomotor or cognitive function.

Generally, the study is well-written, the methods are for the most part appropriate and clearly explained, and the findings make an important contribution to the literature, by investigating for the first time associations between indoor air quality and cognition in the first two years of life. As such, the study may be of broad interest and have an impact, particularly within the fields of neurocognition, child development, and environmental and public health. At the same time, some of the study conclusions are difficult to evaluate based on the information provided. In particular, little information is presented regarding how the study variables relate to each other (and over time) in this sample, why this specific recruitment strategy was chosen, and the rationale for the models tested, particularly the uneven use of covariates. Key strengths and areas for improvement are described in more detail below.

• A key strength of the study is the assessment of indoor air quality – measured repeatedly using an objective device – which may be particularly relevant during the first year of life when children spend much time at home. The authors focused specifically on the presence of very small particles (PM2.5), given previous evidence that these can have neurotoxic and adverse health effects, particularly at high levels of exposure. Levels of indoor air quality were found to largely mirror those of outdoor air quality (based on nearby monitoring stations). This Non-Western sample showed on average exposure to 'unhealthy' levels of PM2.5, based on the US Air Quality Index. The authors found that indoor air quality was linked to meal preparation, and was poorest in homes where solid cooking materials were used. Although it is difficult to draw causal inferences based on these data and to tease apart the effect of cooking materials from broader socio-economic factors, results suggest that the type of cooking fuel used may be an important modifiable target for the improvement of indoor air quality, and potentially downstream health effects.

• Another key strength of the study is the focus on cognition as an outcome during the first two years of life, a period marked by rapid brain growth. It is, however, unclear what the specific rationale was for including children at baseline who were either 6 or 9 months old (e.g., convenience sampling or scientifically motivated), and whether any differences in associations were observed between these two age groups. It is also unclear whether the visual cognition task was developed for this particular study or has been used in previous research, and if so, whether it has been found to be predictive of later cognitive function (including aspects other than visual cognition). The strategy for covariate adjustment appeared to be inconsistent across models tested, with, for example, gender being included as a covariate in some models but not others. In their models, the authors assess two-, three- and four-way interactions in relation to their cognitive outcomes, but no calculations are made to test how powered these analyses are based on the available sample size. Finally, the authors conclude that poor air quality associates with impaired visual cognition across the first two years of age. However, the findings show that for working memory, this association holds true only at baseline, and seems to go in the opposite direction at follow-up. The authors mention that this may be due to the task being too difficult at follow-up, but this is difficult to evaluate given that longitudinal associations between performance in Year 1 and Year 2 are not presented. The authors also mention as an alternative explanation that as children grow up they may spend more time outdoors, so that outdoor air quality may become more important. This may seem unlikely given that indoor and outdoor air pollution were found to largely mirror one another. A more detailed discussion of these findings, as well as potential strategies that could be used in the future to address developmentally-dynamic associations between air quality and cognitive functioning, are warranted.

Below are some specific comments and suggestions that could help improve the manuscript:

– The authors report differences in performance between 6 and 9-month-old infants. It is unclear whether these age-related effects of change preference and memory load are also observed one year later.

– How correlated were the measures of cognition between baseline and follow-up? Generally, I would strongly recommend adding a correlation table/plot with the study variables and covariates, to show how strongly related these are across the type of variable and time point examined.

– Line 95: 'In addition, there was an interaction between SES, measured as a continuous variable based on the Kuppuswamy scales, and year on shift rate, with an increase in shift rate in year 2 of the study, but only for the low SES participants who had a lower starting shift rate in year 1'. This is quite a dense sentence and it would be useful to break it down to explain more clearly the interaction and what it means.

– When examining the association between air quality and cognitive outcomes, why is age cohort adjusted for rather than used as a variable of interest? Is it because the authors assume that associations will be independent of age (or are only interested in testing age-independent effects)?

– The choice of covariates is a bit unclear to me. In particular, gender seems to be included in some models but not in others. It would be helpful to frontload the strategy for covariate adjustment at the beginning of the analysis section.

– Line 178: 'These findings indicate that poor air quality (PM2.5) is associated with impaired visual cognition in the first two years of life'. This sentence seems to only partially reflect the results of the study, as my understanding is that associations with shift rate were only found during year 1, and actually associations show an opposite direction in year 2.

– Line 197: 'It may also indicate that in the second year of life, children spend less time indoors, with outdoor air quality becoming a more important factor'. This sentence seems at odds with the authors' finding that indoor air quality largely mirrored outdoor air quality.

– Figure 3A: it remains unclear to me why associations go in opposite directions between year 1 and year 2, it would be helpful if the authors could expand on this.

– Figure 3B: Why is there a single line here? does it represent year 1 or year 2 of the assessment?

– Table 1: Why is income specifically shown instead of the composite SES variable used in the analyses? Generally, it would be helpful to show percentages to aid the interpretation of the distribution of cooking methods across SES categories.

– Table 5: Did cooking fuel associate with air quality reading even after adjusting for SES? It seems based on this table that cooking fuel and SES are examined as predictors in separate models, so unclear to what extent cooking fuel independently associates with air quality.

*Reviewer #2 (Recommendations for the authors):*

This important study demonstrates that poor indoor air quality impairs cognitive development as early as 6 months of age. The evidence presented is highly compelling, with longitudinal tracking of multiple cognitive indices in a large sample of infants from families across diverse SES strata. The findings have a high public health significance as they support policy changes that will push cooking technologies toward zero emission methods.

1. The authors state that they 'focused on the 'first-look same' trials' as a key aspect of infants' looking is whether they can release fixation from the 'same' side and show a preference for looking at the 'change' side (see 22)" (Line 343-344). This step is unclear: did they only keep these trials in their change preference analyses? If yes, that would be a very unusual analytical choice. The paper that is referenced here (Perone et al., 2012) does not apply this method.

2. Average change preference scores (proportion looking to the changing side) seem to be generally below 0.5, which is puzzling, since the main prediction, based on previous studies, is that there will be an above-chance preference for the changing side. In the current study, this measure varies between chance level performance in the low-load condition to below chance level in the high-load condition (Figure 1D).

3. Air quality is negatively correlated with change preference scores in young infants (Year 1), but not 12 months later (Year 2), which is puzzling. The authors offer a potential explanation, namely that by increasing the set sizes in the task, they inadvertently made it too hard for 18-21-month-olds. This could easily be tested in an exploratory analysis of a subsample of the Year 2 data that is limited to set sizes 2 and 4 only. If the main effect of AQI is not significant, then this would make this explanation unlikely.

4. There was no significant association between air quality and parent-based measures of cognitive skills (MSEL, ASQ) at either of the two-time points. The authors argue that this would be "a possible indication of the need to use specially-designed tasks" (Line 174-175). I do not think this interpretation is warranted. SES had a highly significant relationship with both parent-based scores and age had an effect on ASQ, that is, these measures corroborate previous findings, while these were not so robustly present for the visual WM task measures.

5. The rationale presented for the second dependent variable (shift rate) in the WM study needs to be explained in more detail. If infants are able to detect the changes happening on a particular side, we would expect their gaze to stay longer on that side (which is the basis of the first dependent variable, change preference score). Thus, the two measures are not independent. A recent infant study found that longer looking times (and lower shift rates) reflected higher confidence in a particular choice (Dautriche et al., 2022, Psych Sci).

6. The analysis choice mentioned in #1 needs to be clarified and justified.

7. Conclusions about changes in working memory performance need to be qualified with the note that overall performance did not show the expected level.

8. The alternative explanation mentioned in #3 should be tested and results and interpretation need to be added.

9. The conclusion offered about the relative merits of lab-based and parent-based measures needs to be rephrased.

10. The rationale for the shift rate measure needs to be expanded.

*Reviewer #3 (Recommendations for the authors):*

This paper describes an extensive study testing the impact of air quality, measured in households in Shivgarh, India, on cognitive development during the first two years of life. The size of the cohort, longitudinal design, SES controls, and fine-grained measures of indoor air quality are all strengths of the study. As well, the range of air quality variation and variety of cognitive measures make for a sensitive design. Overall, a picture emerges of a significant, negative association between air quality and cognitive development. However, given the lack of robustness of some of the findings, there are opportunities for further study and discussion.

At the outset, the paper frames the impact of negative neurocognitive health outcomes in broad economic terms. It would be informative to also have a more local, person-centered context that includes individual, family, and community perspectives on the air quality itself, its causes and consequences, and its impact beyond strict economics.

Given the longitudinal design, it would have been interesting to have seen more within-subjects comparisons, with perhaps an exploration of within- versus between-subject variability, individual differences, and their relationship to potential mechanisms of susceptibility, and resilience, to the effects of air pollution.

The authors' concern that the Mullen and ASQ might not be a fully valid measure in this community seems reasonable (though this point could use further elaboration). This concern is used as an argument to explain why there was no significant relationship found between air quality and these measures. However, if I am understanding the tables correctly, both measures were, as would be generally expected, associated with SES. This would tend to actually lend them some support as valid measures in this context. This seems then an apparent discrepancy that the authors could try to reconcile or provide further insight into.

While the authors report significant results showing negative perceptual/cognitive outcomes associated with poor air quality (which is plausible and supported by previous literature and animal work) effect sizes are relatively small and the relationship is not consistent (air quality associated only with 'change preference' in year 1 and 'switch rate' only in year 2, e.g.). Further, there was some treatment of the data that felt arbitrary and made the reader crave both a fuller contextualization and fuller reporting, of the results. Most notably the choice to focus analyses only on the half of the trials where infants had their first fixation on the 'same' (unchanging) stimulus stream. Unless I missed it, it would be nice to see a comparison looking time when the infants have their first look at the changing stimulus stream. Also, analyses were limited to a particular temporal window within a trial, 1500-6500ms. How does this choice influence the results? One way to capture all the richness of the data in one visualization would be to plot 'time course' analyses, showing, ms by ms, the average proportion of time participants spent on the changing versus 'same' streams, perhaps broken down by the stream they start on.

The 'change detection' visual tests used here are common in the field. However, there is no consensus on the mechanisms that they tap into. Here they are described as tests of working memory but given their rapid pace and simple manipulation of basic visual attributes (i.e. noting flickering color changes), many researchers think they instead test earlier, 'lower-level' perceptual/attentional mechanisms. As the authors and others seek, perhaps, to connect air quality to the disruption of particular brain mechanisms and processes (and better relate those mechanisms and processes to subsequent cognitive and educational outcomes), a more precise understanding of the tests themselves will be necessary.

I think most of my comments are covered above. Mostly, I would like to see a fuller description of the results (for visual tests), and graphs could use revising for clarity.

---

## [Author Response]

Essential revisions:1) Please provide more information about the following general aspects of the study methodology and variables examined:a. A rationale for the recruitment strategy and sample selection (e.g. why the study included two cohorts of infants, aged specifically 6 and 9 months of age);

This information has been added (p.3), clarifying that we were expecting a difference in visual working memory (VWM) performance between 6 and 9 months based on studies of western infants. Alternatively, a lack of difference between 6 and 9 months might suggest a developmental delay in VWM in rural India.

b. Adding a table or correlation plot (e.g. in supplementary materials) showing how the study variables relate to each other and over time;

We have added this to the supplementary materials as requested (see Figure S1).

c. Clarifying the strategy for model building and covariate adjustment, as certain covariates (e.g. gender) seem to be included in models unevenly ;

We have clarified our strategy (see p.10-11). Note that we inadvertently included gender in the previous version of the paper based on older analyses of the Mullen/ASQ data; however, we double-checked this and there is no strong statistical reason for including gender. Thus, gender has now been removed making our analysis approach consistent across models.

d. Commenting on the power to test 2-, 3- and 4-way interactions given the available sample size;

We have more carefully specified our analysis strategy (see p.11) and how we tried to construct both theoretically-motivated statistical models and streamlined models that had sufficient power. We address these points in more detail below.

e. Expanding on findings regarding within-subject comparisons.

We have added several follow-up analyses to the results (see p. 6-7) to clarify the findings. This was in response to the reviewer queries below which we discuss in the following sections.

2) The treatment and analysis of data from the computerized tasks need further clarification, particularly in relation to the following points:a. Can the authors clarify why they focused only on 'first-look same trials' (i.e. when infants focused on the unchanging stimulus) in their change preference analyses? As suggested by the reviewers, it would be important to provide a comparison by reporting looking time when infants had their first look at the changing stimulus;

We have clarified why we focused on the ‘first-look no-change’ trials (p. 3). Briefly, a forthcoming paper that examines this sample along with a longitudinal sample of infants from the UK (over 400 infants in the combined sample) shows that the traditional measure of performance in this task – the change preference score – is not stable longitudinally (see p.11). By contrast, our new ‘first-look’ measures are stable longitudinally, that is, year 1 scores measured at 6 and 9 months predict year 2 scores measured at 18 and 21 months within-subjects. Moreover, additional analyses from the same forthcoming paper suggest that ‘first-look no-change’ scores are most indicative of VWM capacity. In this context, it is interesting that we find associations between air quality and the ‘first-look no-change’ scores only. Note that we have also added analyses of ‘first-look change’ scores in the revised manuscript (see Tables 10 and 11). There were no associations between this second measure and air quality.

b. Analyses were limited to a specific temporal window, and it is unclear how this choice might have influenced the results. Can the authors provide more information about this, for example using the 'time course' plot suggested by Reviewer 3;

We have added more rationale for the temporal window chosen and refined this window a bit more in the revised paper (see p.11).

c. The authors should comment on potential reasons why change preference scores in this study were on average lower than 0.5 (i.e. chance level) and how this might impact the interpretation of findings;

We have clarified (p.3) that the new ‘first-look no-change’ measure is not anchored at 0.5 (since, conceptually, this measure starts at 0 and moves away from 0 over the time window).

d. The rationale for using the shift rate measure needs to be expanded given the potential for inter-dependence with change preference score.

We have expanded the rationale here (p. 3). We have also added a follow-up test to show that both associations with air quality (i.e., ‘first-look no-change’ to air quality and shift rate to air quality) are robust, even when the complementary measure is added to the model. For instance, air quality continues to be associated with the ‘first-look no-change’ measure even when shift rate is added to the statistical model (see p.6).

3) Re-evaluating and expanding the interpretation of findings:a. The authors mention that the lack of significant associations between indoor air pollution and questionnaire-based measures of cognition may be due to the measures having lower validity for this type of sample. However, as pointed out by several reviewers, these measures do associate with other study variables, such as SES, as expected. Can the authors revise their conclusions accordingly and expand on alternative potential reasons for the lack of associations with indoor air pollution?

We have revised this section and thank the reviewers for the comment. Our conclusion is that air quality has a specific impact on the visual cognitive system. This is an early-developing system that is indexed by the preferential looking task. By contrast, the questionnaire-based measures target more general aspects of development. Thus, air quality appears to impact VWM specifically rather than these more general aspects of development. Note that we cannot conclude that VWM is the only system impacted by air quality. It is possible that other early-developing systems are also impacted; this will require further study. We mention, for instance, that it would be interesting to probe if air quality impacts auditory and/or statistical learning processes (p. 8).

b. Conclusions about associations between indoor air pollution and performance on the visual cognition task should be qualified by stating more clearly that observed effect sizes were generally small, the relationship was not consistent over time and that overall performance on change preference was lower than expected (i.e. on average below 0.5). As suggested by Reviewer 2, the authors should also perform a post hoc analysis to test whether an association between air quality and change preference scores in Year 2 is observed when restricting data to set sizes 2 and 4 only. This will help readers evaluate whether the lack of associations in Year 2 may be due to the task being too difficult or whether other factors may be at play.

We have qualified our conclusions remarking on the effect sizes and the inconsistency for the VWM scores in year 2. On that front, we note that other data from our study suggest that VWM in year 2 is impacted by interactions with the caregiver; thus, it is possible that air quality is primarily impactful early in infancy when this cognitive system is initially organizing but other factors play a more central role as the system changes from year 1 to year 2 (see p. 8-9).

We did note that the change preference scores were below 0.5 and that this was expected with the new ‘first-look no-change’ measure.

Finally, we ran the suggested analysis including only the ‘load 2’ data. This is useful in that we still get a year x air quality interaction. Thus, it does not appear that the VWM task is too difficult in year 2 to detect an effect.

Reviewer #1 (Recommendations for the authors):In this study, Spencer et al. examine whether indoor air quality is associated with multiple aspects of infant cognitive functioning, based on longitudinal data from a sample of families from rural India. Briefly, infants were enrolled at either 6 or 9 months of age. Laboratory assessments were performed at baseline and repeated one year later, whereas home visits to measure air quality took place after each laboratory assessment (i.e., baseline and follow-up), and were repeated every three months for a period of one year. The main effects of air quality as well as interactions with factors such as age, gender, and socio-economic status were tested. Based on these data, the authors report an association between poorer air quality and worse performance on a computerized task of visual cognition. Specifically, associations with processing speed were observed at both time points, while associations with working memory were only observed at baseline (year 1). No associations were identified with standardized questionnaire-based measures of psychomotor or cognitive function.Generally, the study is well-written, the methods are for the most part appropriate and clearly explained, and the findings make an important contribution to the literature, by investigating for the first time associations between indoor air quality and cognition in the first two years of life. As such, the study may be of broad interest and have an impact, particularly within the fields of neurocognition, child development, and environmental and public health. At the same time, some of the study conclusions are difficult to evaluate based on the information provided. In particular, little information is presented regarding how the study variables relate to each other (and over time) in this sample, why this specific recruitment strategy was chosen, and the rationale for the models tested, particularly the uneven use of covariates. Key strengths and areas for improvement are described in more detail below.• A key strength of the study is the assessment of indoor air quality – measured repeatedly using an objective device – which may be particularly relevant during the first year of life when children spend much time at home. The authors focused specifically on the presence of very small particles (PM2.5), given previous evidence that these can have neurotoxic and adverse health effects, particularly at high levels of exposure. Levels of indoor air quality were found to largely mirror those of outdoor air quality (based on nearby monitoring stations). This Non-Western sample showed on average exposure to 'unhealthy' levels of PM2.5, based on the US Air Quality Index. The authors found that indoor air quality was linked to meal preparation, and was poorest in homes where solid cooking materials were used. Although it is difficult to draw causal inferences based on these data and to tease apart the effect of cooking materials from broader socio-economic factors, results suggest that the type of cooking fuel used may be an important modifiable target for the improvement of indoor air quality, and potentially downstream health effects.• Another key strength of the study is the focus on cognition as an outcome during the first two years of life, a period marked by rapid brain growth. It is, however, unclear what the specific rationale was for including children at baseline who were either 6 or 9 months old (e.g., convenience sampling or scientifically motivated), and whether any differences in associations were observed between these two age groups.

Thanks for the comment. We have added a more detailed rationale for these ages in the revised manuscript. Briefly, there is a change in VWM capacity between 6 and 8.5 months; thus, we divided our cohort into two groups on either side of this developmental transition. This allowed us to probe how VWM changes over ages in rural India relative to western samples. Note that we included Age as a predictor in all models and did not see any evidence that the air quality associations reported here were related to Age.

It is also unclear whether the visual cognition task was developed for this particular study or has been used in previous research, and if so, whether it has been found to be predictive of later cognitive function (including aspects other than visual cognition).

The preferential looking task has been used in prior work, but, until recently, there have not been any longitudinal assessments. We have now completed two longitudinal studies – one in the UK and one in rural India. This work motivated the particular measures used here (for discussion, see p.11).

The strategy for covariate adjustment appeared to be inconsistent across models tested, with, for example, gender being included as a covariate in some models but not others.

Apologies for the oversight regarding gender. We have now carefully re-run all analyses and found no evidence warranting the inclusion of gender. Thus, this predictor has been removed from all analyses.

In their models, the authors assess two-, three- and four-way interactions in relation to their cognitive outcomes, but no calculations are made to test how powered these analyses are based on the available sample size.

In all models, we aimed to theoretically motivate the inclusion of all effects and limit spurious correlations by only adding effects where they were thought to contribute a priori. In the case of interactions, models were tested to see whether interactions improved model fit through formal model comparison, and interactions were not included if they did not contribute to model fit. The goal was to create the most streamlined models possible to ensure there was sufficient power in all analyses.

Finally, the authors conclude that poor air quality associates with impaired visual cognition across the first two years of age. However, the findings show that for working memory, this association holds true only at baseline, and seems to go in the opposite direction at follow-up. The authors mention that this may be due to the task being too difficult at follow-up, but this is difficult to evaluate given that longitudinal associations between performance in Year 1 and Year 2 are not presented.

We have conducted longitudinal analyses in a forthcoming paper, combining the current data set with data from a longitudinal study in the UK (N > 400 total). Results revealed that the standard measure from the preferential looking task – the change preference score – was not stable longitudinally. Critically, the new ‘first look’ measures used here ARE stable longitudinally. We provide an overview of these findings on p. 11. In addition, we have added a correlation table to the supplementary materials so the reader can get a sense for how the key measures reported here are associated.

We also appreciate the comment about task difficulty. We added a new follow-up analysis to the paper comparing load 2 for year 1 and year 2. Results show that the Year x AQI interaction is robust with this dataset, ruling out the task difficulty explanation. We have updated the Discussion accordingly. Note that we suspect there is no robust link between change preference scores and air quality in year 2 because other factors which are not correlated with air quality are impacting the development of VWM. We mention one possibility (dyadic interactions with the caregiver) on p. 8.

The authors also mention as an alternative explanation that as children grow up they may spend more time outdoors, so that outdoor air quality may become more important. This may seem unlikely given that indoor and outdoor air pollution were found to largely mirror one another. A more detailed discussion of these findings, as well as potential strategies that could be used in the future to address developmentally-dynamic associations between air quality and cognitive functioning, are warranted.

This is a fair point. We have removed the comment about outdoor air quality in the revised paper and have added a more detailed discussion of our findings (p. 7-8).

Below are some specific comments and suggestions that could help improve the manuscript:– The authors report differences in performance between 6 and 9-month-old infants. It is unclear whether these age-related effects of change preference and memory load are also observed one year later.

The age effect in Table 2 is an Age main effect; this suggests there is some common variance that extends over both years in the study. Similarly, the Load effect is a main effect. We did not find evidence that Age or Load interacted with other variables.

– How correlated were the measures of cognition between baseline and follow-up? Generally, I would strongly recommend adding a correlation table/plot with the study variables and covariates, to show how strongly related these are across the type of variable and time point examined.

We have added a correlation table to the supplementary materials as requested. We also note on p. 11 that the ‘first-look’ measures we used here are stable longitudinally (as we describe in a forthcoming paper).

– Line 95: 'In addition, there was an interaction between SES, measured as a continuous variable based on the Kuppuswamy scales, and year on shift rate, with an increase in shift rate in year 2 of the study, but only for the low SES participants who had a lower starting shift rate in year 1'. This is quite a dense sentence and it would be useful to break it down to explain more clearly the interaction and what it means.

In cleaning up our analysis script, we noticed that some missing observations had crept into this data frame. Consequently, the Year x SES interaction is no longer significant and this dense sentence has been deleted.

– When examining the association between air quality and cognitive outcomes, why is age cohort adjusted for rather than used as a variable of interest? Is it because the authors assume that associations will be independent of age (or are only interested in testing age-independent effects)?

Each model was conceived with parsimony in mind; we also sought to simplify the models based on theoretical motivations. We adjusted for age because we theorise that the older cohort might have a higher baseline cognitive score, but we had no reason to think that this difference would change over time as a function of air quality. We note that we did explore possible interactions with Age in initial models and found no robust relationships.

– The choice of covariates is a bit unclear to me. In particular, gender seems to be included in some models but not in others. It would be helpful to frontload the strategy for covariate adjustment at the beginning of the analysis section.

Apologies for this lack of consistency. As described above, this has been cleaned up. In short, we found no evidence of links between our measures and gender.

– Line 178: 'These findings indicate that poor air quality (PM2.5) is associated with impaired visual cognition in the first two years of life'. This sentence seems to only partially reflect the results of the study, as my understanding is that associations with shift rate were only found during year 1, and actually associations show an opposite direction in year 2.

We have revised this sentence: “These findings indicate that poor air quality (PM_2.5_) is associated with slower visual processing speed in the first two years of life and poorer visual working memory scores in year 1.”

– Line 197: 'It may also indicate that in the second year of life, children spend less time indoors, with outdoor air quality becoming a more important factor'. This sentence seems at odds with the authors' finding that indoor air quality largely mirrored outdoor air quality.

That is correct and we have edited the discussion to more carefully reflect this point.

– Figure 3A: it remains unclear to me why associations go in opposite directions between year 1 and year 2, it would be helpful if the authors could expand on this.

We have enriched our Discussion of this result on p.7.

– Figure 3B: Why is there a single line here? does it represent year 1 or year 2 of the assessment?

The analyses show a main effect of shift rate; thus, we have aggregated shift rate over years to reflect this main effect.

– Table 1: Why is income specifically shown instead of the composite SES variable used in the analyses? Generally, it would be helpful to show percentages to aid the interpretation of the distribution of cooking methods across SES categories.

Income is useful here to show the distribution of income relative to SES (which is a composite of income, education, occupation). As requested, we have converted the data to proportions.

– Table 5: Did cooking fuel associate with air quality reading even after adjusting for SES? It seems based on this table that cooking fuel and SES are examined as predictors in separate models, so unclear to what extent cooking fuel independently associates with air quality.

Thanks for this comment. SES and cooking fuel choice are highly correlated (see Table 1) and theoretically should be as well, so this violates the assumption of independence in linear models. As such, we represent them as two models here to highlight their correspondence.

Reviewer #2 (Recommendations for the authors):This important study demonstrates that poor indoor air quality impairs cognitive development as early as 6 months of age. The evidence presented is highly compelling, with longitudinal tracking of multiple cognitive indices in a large sample of infants from families across diverse SES strata. The findings have a high public health significance as they support policy changes that will push cooking technologies toward zero emission methods.1. The authors state that they 'focused on the 'first-look same' trials' as a key aspect of infants' looking is whether they can release fixation from the 'same' side and show a preference for looking at the 'change' side (see 22)" (Line 343-344). This step is unclear: did they only keep these trials in their change preference analyses? If yes, that would be a very unusual analytical choice. The paper that is referenced here (Perone et al., 2012) does not apply this method.

We have expanded our discussion of how we treat the change preference scores. The new additional discussion hopefully clarifies these points. Note that we cited Perone et al. here as we have used our neurocomputational model to understand the new ‘first-look’ measures described herein, but this citation has now been removed for clarity.

2. Average change preference scores (proportion looking to the changing side) seem to be generally below 0.5, which is puzzling, since the main prediction, based on previous studies, is that there will be an above-chance preference for the changing side. In the current study, this measure varies between chance level performance in the low-load condition to below chance level in the high-load condition (Figure 1D).

As described above and in the revised paper, the new ‘first-look’ measures we used are not anchored around 0.5 (because ‘first-look no-change’ starts at 0 and ‘first-look change’ starts at 1). We have clarified this in the revised paper.

3. Air quality is negatively correlated with change preference scores in young infants (Year 1), but not 12 months later (Year 2), which is puzzling. The authors offer a potential explanation, namely that by increasing the set sizes in the task, they inadvertently made it too hard for 18-21-month-olds. This could easily be tested in an exploratory analysis of a subsample of the Year 2 data that is limited to set sizes 2 and 4 only. If the main effect of AQI is not significant, then this would make this explanation unlikely.

This was an excellent suggestion. We conducted this analysis and the year X AQI interaction remains. Thus, this is not a likely explanation for the effect.

4. There was no significant association between air quality and parent-based measures of cognitive skills (MSEL, ASQ) at either of the two-time points. The authors argue that this would be "a possible indication of the need to use specially-designed tasks" (Line 174-175). I do not think this interpretation is warranted. SES had a highly significant relationship with both parent-based scores and age had an effect on ASQ, that is, these measures corroborate previous findings, while these were not so robustly present for the visual WM task measures.

Yes. Fair play. Perhaps the most direct explanation is that air quality impacts VWM but not other more general aspects of motor/cognitive/social development. That is, maybe the impact on VWM is specific and the test that assesses VWM picks up on this. We have expanded the discussion to highlight these points.

5. The rationale presented for the second dependent variable (shift rate) in the WM study needs to be explained in more detail. If infants are able to detect the changes happening on a particular side, we would expect their gaze to stay longer on that side (which is the basis of the first dependent variable, change preference score). Thus, the two measures are not independent. A recent infant study found that longer looking times (and lower shift rates) reflected higher confidence in a particular choice (Dautriche et al., 2022, Psych Sci).

This is a good point. We had a priori reasons to look at both measures. That said, we conducted two follow-up analyses to see if some shared variance among measures was driving our results. These analyses suggest that links between air quality and the visual cognition measures are robust, even when controlling for this shared variance by including both visual cognition measures in the same models.

A recent infant study found that longer looking times (and lower shift rates) reflected higher confidence in a particular choice (Dautriche et al., 2022, Psych Sci).

There is a long history of interpreting shift rates in the infant literature. While the cited paper is interesting, we opted not to include this in the revised paper, instead interpreting shift rates in line with our prior neurocomputational models (see, e.g., Perone et al., 2012).

6. The analysis choice mentioned in #1 needs to be clarified and justified.

Yes. We have expanded our discussion of the analyses. See, for instance, p. 10-11.

7. Conclusions about changes in working memory performance need to be qualified with the note that overall performance did not show the expected level.

We have clarify this point above and in the revised paper. Our new ‘first look’ measures are not expected to be anchored to 0.5.

8. The alternative explanation mentioned in #3 should be tested and results and interpretation need to be added.

Done. See above.

9. The conclusion offered about the relative merits of lab-based and parent-based measures needs to be rephrased.

This has been updated.

10. The rationale for the shift rate measure needs to be expanded.

This has been added.

Reviewer #3 (Recommendations for the authors):This paper describes an extensive study testing the impact of air quality, measured in households in Shivgarh, India, on cognitive development during the first two years of life. The size of the cohort, longitudinal design, SES controls, and fine-grained measures of indoor air quality are all strengths of the study. As well, the range of air quality variation and variety of cognitive measures make for a sensitive design. Overall, a picture emerges of a significant, negative association between air quality and cognitive development. However, given the lack of robustness of some of the findings, there are opportunities for further study and discussion.At the outset, the paper frames the impact of negative neurocognitive health outcomes in broad economic terms. It would be informative to also have a more local, person-centered context that includes individual, family, and community perspectives on the air quality itself, its causes and consequences, and its impact beyond strict economics.

We have added a more family-centred perspective in the Introduction and Discussion, highlighting the impact air quality can have on families, particularly via developmental impacts which can lead to emotional and behavioural problems.

Given the longitudinal design, it would have been interesting to have seen more within-subjects comparisons, with perhaps an exploration of within- versus between-subject variability, individual differences, and their relationship to potential mechanisms of susceptibility, and resilience, to the effects of air pollution.

All of our linear mixed-effects models were within-subjects longitudinally; thus, we are not sure exactly what the reviewer would like to see here. In the revised paper, we tried to clarify our analysis approach. We also note that we used measures that were longitudinally stable (see, e.g., the discussion of our ‘first-look’ measures on p. 11).

The authors' concern that the Mullen and ASQ might not be a fully valid measure in this community seems reasonable (though this point could use further elaboration). This concern is used as an argument to explain why there was no significant relationship found between air quality and these measures. However, if I am understanding the tables correctly, both measures were, as would be generally expected, associated with SES. This would tend to actually lend them some support as valid measures in this context. This seems then an apparent discrepancy that the authors could try to reconcile or provide further insight into.

This is an excellent point. We have refined our discussion of the Mullen/ASQ findings. We think our results suggest that visual cognition is specifically impacted by air quality (vs more general aspects of cognition/motor/social functioning).

While the authors report significant results showing negative perceptual/cognitive outcomes associated with poor air quality (which is plausible and supported by previous literature and animal work) effect sizes are relatively small and the relationship is not consistent (air quality associated only with 'change preference' in year 1 and 'switch rate' only in year 2, e.g.).

We have noted the relatively small effect sizes. We also note that effects of air quality on shift rate were robust across both years.

Further, there was some treatment of the data that felt arbitrary and made the reader crave both a fuller contextualization and fuller reporting, of the results. Most notably the choice to focus analyses only on the half of the trials where infants had their first fixation on the 'same' (unchanging) stimulus stream.

This is an excellent suggestion. We have added these analyses, noting no relationships between air quality and the ‘first look change’ trials.

Unless I missed it, it would be nice to see a comparison looking time when the infants have their first look at the changing stimulus stream. Also, analyses were limited to a particular temporal window within a trial, 1500-6500ms. How does this choice influence the results? One way to capture all the richness of the data in one visualization would be to plot 'time course' analyses, showing, ms by ms, the average proportion of time participants spent on the changing versus 'same' streams, perhaps broken down by the stream they start on.

This is an excellent suggestion; however, this is precisely what we do in a forthcoming paper examining the current dataset combined with a large longitudinal study in the UK (N > 400). Rather than repeat this analysis here, we refer to our forthcoming paper providing an overview of our findings.

The 'change detection' visual tests used here are common in the field. However, there is no consensus on the mechanisms that they tap into. Here they are described as tests of working memory but given their rapid pace and simple manipulation of basic visual attributes (i.e. noting flickering color changes), many researchers think they instead test earlier, 'lower-level' perceptual/attentional mechanisms. As the authors and others seek, perhaps, to connect air quality to the disruption of particular brain mechanisms and processes (and better relate those mechanisms and processes to subsequent cognitive and educational outcomes), a more precise understanding of the tests themselves will be necessary.

This is an excellent point and is something we are working on. For instance, in our forthcoming paper, we use the neurocomputational model from Perone et al. (2012) to explain how the ‘first-look’ measures are influenced by individual differences in VWM. Inclusion of such work here is beyond the scope of the present paper, but we do point to our forthcoming paper on this topic.